# Participatory Rural Spatial Planning Based on a Virtual Globe-Based 3D PGIS

**Linjun Yu [1], Xiaotong Zhang [2], Feng He [3,*], Yalan Liu [1] and Dacheng Wang [1]**

[1]  Aerospace Information Research Institute, Chinese Academy of Sciences, Beijing 100094, China;
    yulj201831@aricas.ac.cn (L.Y.); liuyl@aircas.ac.cn (Y.L.); wangdc@aircas.ac.cn (D.W.)

[2]  China National Engineering Research Center for Human Settlements, Beijing 100044, China;
    zhangxt@cadg.cn

[3]  School of Urban and Environment, Yunnan University of Finance and Economics, Kunming 650221, China

[*]  Correspondence: zz1923@ynufe.edu.cn; Tel.: +86-13629431746

**Abstract:** With the current spatial planning reform in China, public participation is becoming increasingly important in the success of rural spatial planning. However, engaging various stakeholders in spatial planning projects is difficult, mainly due to the lack of planning knowledge and computer skills. Therefore, this paper discusses the development of a virtual globe-based 3D participatory geographic information system (PGIS) aiming to support public participation in the spatial planning process. The 3D PGIS-based rural planning approach was applied in the village of XiaFan, Ningbo, China. The results demonstrate that locals' participation capacity was highly promoted, with their interest in 3D PGIS visualization being highly activated. The interactive landscape design tools allow stakeholders to present their own suggestions and designs, just like playing a computer game, thus improving their interactive planning abilities on-site. The scientific analysis tools allow planners to analyze and evaluate planning scenarios in different disciplines in real-time to quickly respond to suggestions from participants on-site. Functions and tools such as data management, marking, and highlighting were found to be useful for smoothing the interactions among planners and participants. In conclusion, virtual globe-based 3D PGIS highly supports the participatory rural landscape planning process and is potentially applicable to other regions.

**Keywords:** participatory rural spatial planning; public participation; participatory geographic information system; virtual globe; 3D visualization

## 1. Introduction

Small towns and villages play a crucial role in landscape, ecological service, and environmental protection. According to the Town Planning Standard of China (GB50188-2007), "small town" refers to towns with a population of less than 10,000, and a village usually has a population of less than 1000. Although urban and rural areas in China have been experiencing dramatic changes in terms of population migration, land use, landscape, economic growth, and living style since the late 1970s [1–3], coordinated urban–rural development is far from being achieved [1,4]. Unlike urban planning, rural planning is not mandatory, which has led to only a few small towns and villages in China undertaking spatial planning. In some developed regions, government-oriented administrative planning has intervened in rural spatial restructuring, generally imitating the urban mode to construct rural space, resulting in the loss of unique rural characteristics [5,6]. Due to the lack of spatial planning, small towns and villages in rural areas in China are usually spontaneously developing, resulting in a number of problems such as landscape aesthetical issues and ecological and environmental issues. In this context, numerous government policies and development strategies, such as New Countryside Construction and The Beautiful Countryside, have been launched by the Chinese government to save resources and achieve environmentally friendly

rural spatial development in China. Scientifically planning the spatial pattern of villages and small towns has been included in the 13th Five-Year Plan for Economic and Social Development of the People's Republic of China (2016–2020) [7]. Therefore, in the posturbanization process in China, rural spatial planning will face new opportunities and challenges.

Rural spatial planning is different from urban planning, as rural areas have a wide range of opportunities and problems that are unique compared to urban areas. One of the features of rural spatial planning is that it requires more of the locals' participation in the planning process because they are not only the stakeholders directly affected by the consequences of planning but are also a unique source of local knowledge and perspectives.

As spatial planning concerns various stakeholders, such as the general public, locals, regional authorities, and other interested parties, public participation has been a focus of spatial planning and is required by law in many countries such as the United States, the United Kingdom, and Japan [8]. Democracy and participation have been the focus of spatial planning studies for several decades (see, for example, Zhang, et al. [9], Arler [10], Jones [11], Stenseke [12], Caspersen [13], Selman [14] and Stenseke and Jones [15]). Respecting the will of the villagers in a township or village planning is required in the Urban and Rural Planning Law of the People's Republic of China (2019 Amendment) [16]. However, due to the culture and political structure, participatory planning has so far been insufficiently applied in China. Specifically for rural planning, to avoid the villages all looking the same and losing their own unique distinction characteristics, which has already occurred in urban development in China, the participation of local inhabitants and other stakeholders is essential.

The use of participatory planning is an effective means to improve community engagement [17,18]. Various methods and tools have been applied in participatory planning such as surveys, meetings, and mapping. With the development of geographic information systems (GISs), public participation geographic information systems (PPGIS), aiming to use GIS to broaden public involvement in policymaking, have received increasing academic interest and has been applied in the field of planning [19–22]. However, gaps remain between application requirements from on-site interactions in rural spatial planning and the capacity of current PGISs. First, visual representations are particularly important mediators of geocollaborative activities [23]. The 2D map commonly used in current PGISs is insufficient to create an intuitive communication platform among planners and various stakeholders, especially for laypersons who usually cannot transform abstract 2D mapping into intuitive imaginations. However, although 3D visualization has been widely used for final planning presentation in many planning applications (see, for example, Lovett et al. [24] and Paar [25]), there is a shortage of intuitive and interactive 3D platforms supporting the participatory spatial planning process. Second, an easy-to-use PPGIS with which various stakeholders can interact in a simple manner during on-site discussions is urgently required. Third, many scientific assessment tools supporting real-time planning scenarios assessment should be integrated into the PPGIS. Therefore, to address these challenges and demands of participatory rural planning in China, a 3D virtual globe-based PGIS was constructed in this study. In this paper, a framework and solutions to technical issues of developing the prototype system are introduced. In addition, we explore the use of a 3D PGIS-based planning approach in the participatory rural landscape planning project of the village of XiaFan, Ningbo, China, to evaluate this prototype system.

## 2. Related Technologies

Public participation, based on the belief that people affected by a decision own a right to be involved in the decision-making process [26], has been a focus of spatial planning research and practice for several decades [10,22,27]. In practice, participatory planning has covered a wide variety of activities, mainly including information collection and exchange, data analysis, discussion, negotiation, decision making, etc. To achieve efficient public participation, various methods and tools have been applied. Paper-based maps and text specifications are the earliest and still popular methods in participatory planning. With the advances in computer and information technologies, computer-aided

design (CAD) tools have been widely applied to support spatial planning processes by conveniently generating sketches or digital planning maps for visualization and discussion.

In recent years, traditional GISs have become some of the most commonly used tools in spatial planning. However, their intrinsic complex and professional nature limit their use in engaging different groups of stakeholders as a community in the planning process [28,29]. Participatory GIS (PGIS), therefore, was then proposed to bridge the gap between participatory planning and GIS using a simplified graphical user interface (GUI) [28]. The advancement of web-based GIS is leading PGIS implementation in a web-based framework (Web-PGIS), allowing even more nonprofessionals to participate in spatial planning processes and anonymously express their views [30,31].

Visualization of the design and planning sketch is a key component of PGIS. However, the 2D maps and images used in traditional GIS and web GIS are insufficient to provide the public with an intuitive visualization environment. Studies have shown that 3D visualization can facilitate public participation in planning (see, for example, Lovett, et al. [24], Lafrance, et al. [30], Howard and Gaborit [32] and Lloret et al. [33]). Various 3D visualization technologies have been applied to support participation in spatial planning such as visual reality (VR) [27,34], Java 3D [35], virtual reality markup language (VRML) [36,37], the Extensible 3D (X3D) X3D [38], keyhole markup language (KML) [39], and virtual globe-based 3D online visualization [40]. However, these 3D visualization solutions have been generally used to present planning scenarios for specific projects and do not provide an interactive design environment, limiting their impact on public participation [27]. The planner and public would expect a 3D PGIS to be a platform that can provide an interactive 3D design environment, allowing planners and participants to communicate for information collection and interactively design the spatial plan. It should also provide abundant analysis functions allowing planners to instantly and scientifically evaluate the planning scenario, which can improve the communication efficiency among participants. A 3D PGIS should be capable of planning information sharing, allowing the public to be consulted remotely.

In recent years, virtual globe-based 3D visualization technologies have matured and been accepted by the public. Many virtual globe software programs have been released, including commercial software such as Skyline Globe, Google Earth, Microsoft Virtual Earth, and ArcGIS Explorer, as well as open-source software such as National Aeronautics and Space Administration (NASA) World Wind, Cesium, and osgEarth. As a result of their continuous development, virtual globes now have been applied for presenting global geospatial data for public accessing as well as for scientific research and commercial usage in various fields (for example, Huang [35], Wu, et al. [40], Zheng, et al. [41] and Yu and Gong [42]). Virtual globe technology is suitable for creating a 3D environment for public participation [18,30,40]. However, participatory rural planning requires face-to-face interactions, not the online model used in urban planning, as local rural inhabitants in China are usually unfamiliar with computers and have little planning knowledge. Therefore, a participation platform supporting on-site communications is required as many individual demands and conflicts needed to be expressed, discussed, and solved on-site.

## 3. Materials and Methods

### 3.1. The Requirements for 3D PGIS from Participatory Rural Spatial Planning

The definition of the user requirements is the first step in the construction of geoinformation solutions [43]. The objective of this study was to provide a 3D PGIS solution to support participatory rural spatial planning. To provide effective visualization, individual differences must be carefully considered [44]. Three-dimensional (3D) visualization was selected because local villagers, as the major participants in rural spatial planning, usually have little knowledge of planning, and 3D visualization can help them to better understand planning and communicate with planners. Specific feature requirements of the 3D PGIS can be obtained using the typical workflow of participatory rural spatial planning, as shown in Figure 1. Phase 1 is planning preparation, in which basic data and information about

the village are collected, including basic geographic data, previous spatial planning, planning at the town level or above, natural and human resources data, social and economic development data, etc. In this phase, the 3D PGIS acts as an information collection and visualization tool, supporting the creation of the present and historical 3D landscape of the village, which are the base for further planning. The major tasks of Phase 2 include presenting the status analysis; 3D PGIS is used to implement problem identification and demand analysis with the objective of preparing the necessary materials for future discussion in the next phase. In Phase 3, interactive planning and design are implemented using the 3D PGIS, through which various stakeholders can directly communicate in the 3D environment. Therefore, interactive landscape and design tools, marking and highlighting tools, and scientific analysis tools are required. To achieve the objective of effective communication among various participants, this phase requires these tools to be very easy to use, allowing planners to quickly respond to stakeholders' demands and suggestions. Phases 4 and 5 are postplanning phases in which the 3D PGIS acts as an information system for further plan implementation and monitoring.

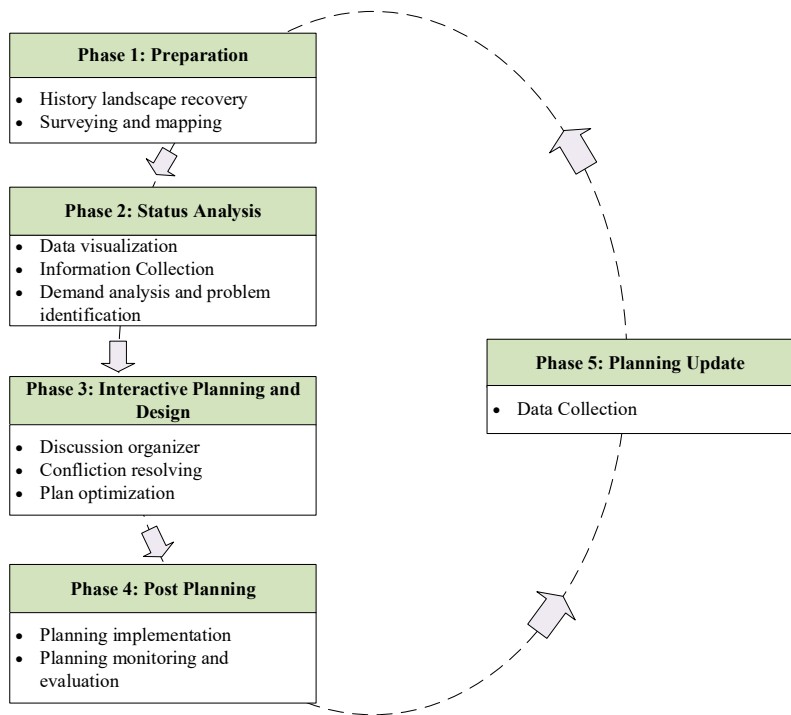

**Figure 1.** The workflow of 3D participatory geographic information system (PGIS)-based participatory spatial planning.

### 3.2. Three-Dimensional (3D) Virtual Globe Platform

The virtual globe-based 3D visualization technology provides a suitable platform for developing public participation systems by providing intuitive 3D visualizations of various spatial data from the global scale to the street scale. As shown in Figure 2, a service-oriented architecture (SOA)-based virtual globe system [45] was used as the platform for the 3D PGIS. On the server side, web services provide data and spatial analysis services to the virtual globe client. As large-scale data, e.g., high-resolution remote-sensing images, must be rendered using level of detail (LOD) algorithms in the client, data are usually initially processed into smaller tiles according to the LOD rule and stored in data caches both on server and client sides. The GIS server on the server side provides GIS analysis functions and tiled data generation functions to the client through web services. Third-party data can also be accessed and rendered in the virtual globe client using the web feature services (WFS) and the web map services (WMS). Based on this framework, application models and functions can be developed. More technical details of this virtual globe system can be found in the work of Yu et al. [45].

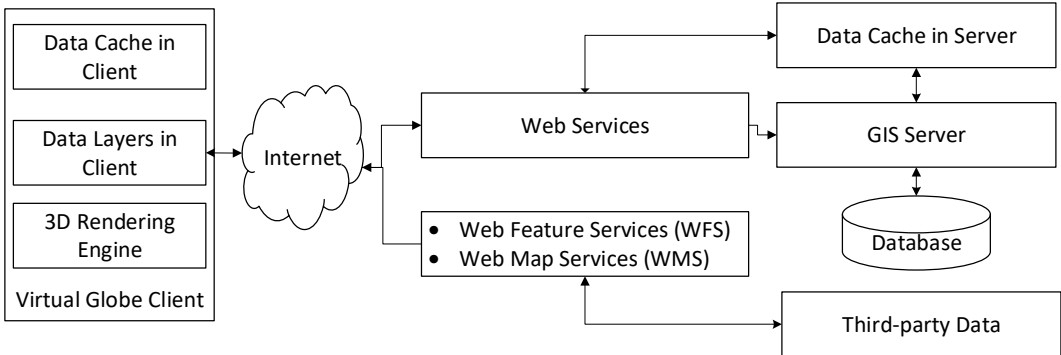

**Figure 2.** The general framework of the 3D virtual globe platform.

### 3.3. General Architecture for Virtual Globe-Based 3D PGIS

The general architecture of the virtual globe-based 3D PGIS for rural spatial planning is shown in Figure 3. The virtual globe is the development platform providing the basic 3D development environment and GIS functions through interfaces such as data interfaces, interactive operation interfaces, and spatial analysis interfaces. Based on the virtual globe-based platform, a series of participatory planning-oriented tools, including interactive landscape design tools, scientific analysis and evaluation tools, and planning and management tools, were developed to support participatory rural spatial planning. The interactive landscape design tools are used to interactively design landscape elements, such as roads, buildings, land parcels, and planning boundaries, in a simple and direct manner. The scientific analysis and evaluation tools are composed of a group of spatial analysis tools, such as skyline analysis, landscape pattern analysis, and ecological service value analysis, to provide scientific analyses and evaluations of planning scenarios. The planning organization tools provide functions such as data management of planning the project, results output, and marking and data highlighting to assist with the organization of the participatory planning process.

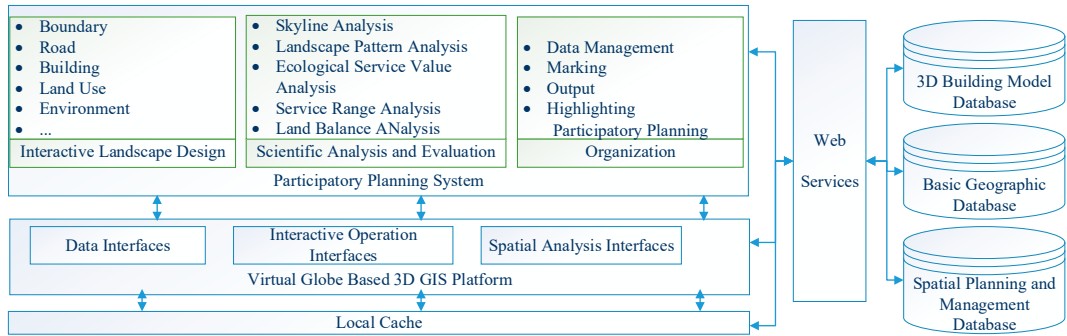

**Figure 3.** The system architecture of the virtual globe-based 3D PGIS for rural spatial planning.

The service side provides data services to the client through web serveries. The 3D building model database stores 3D building models by category. The basic geographic database stores various spatial data, including the digital elevation model (DEM), remote-sensing images, etc. The spatial planning and management database stores spatial planning project data such as project information, planning data, and marks.



*3.4. Key Components*

3.4.1. Interactive Landscape Design Module

Landscape design is one of the major tasks in rural spatial planning. A landscape scene usually consists of a series of landscape objects such as roads, boundaries, and buildings. To facilitate the on-site communication of various stakeholders, the landscape design tools should be simple and easy to use, allowing participants to interactively create and modify landscape objects in the 3D environment just like playing a computer game. However, as the designed landscape is not only used for the visualization of current or historical landscape status or planning vision but also provides the base for subsequent scientific analysis, the landscape data should be well organized. Since landscape elements are overlaid and interrelated in space, changes to a landscape element may affect others. Therefore, a landscape object cannot simply be placed into a 3D scene regardless of its impacts on others. As shown in the framework for interactive landscape design (Figure 4), landscape design tools form a set of interrelated tools, allowing the interactive creation and editing of 3D landscape elements.

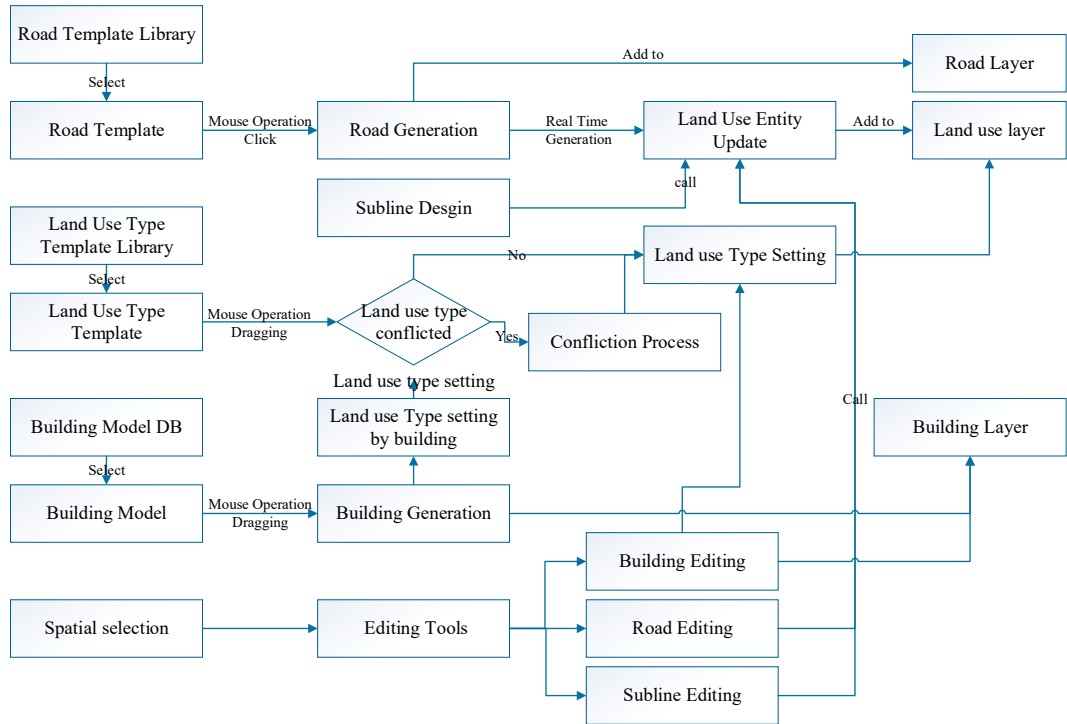

**Figure 4.** The framework of interactive landscape design.

3.4.2. Interactive Road Design

Roads are an important type of landscape element; they function to divide the planning space into different functional regions. Therefore, spatial planning often starts with the road layout, which is one of the core issues concerning multiple participants in the participatory planning process. Therefore, to enable participants to quickly and accurately express their assumptions about the spatial pattern during on-site discussion, a simple and fast road design tool is required.

An interactive road design tool that provides a simple and quick road design for rural spatial planning was developed. As shown in Figure 4, the user first selects a road template, which defines the 3D structure of a road segment from the road template library, and then interactively draws a series of points on the 3D earth to form the road centerline. The road generation method is then used to automatically generate 3D road segments according to the selected road template and the centerline. As roads are also the boundaries of land parcels, the land use updating function, which generates new

land use entities or updates boundaries of existing ones, is then automatically triggered after finishing the operation of road creation or editing.

A road template consists of the following attributes:

$$\text{Road Template} = \{d_{rw}, d_{cw}, d_{sw}, d_{gb}, bHasTrees, strFilePath, bHasLamps, strLampFilefile\}, \qquad (1)$$

where $d_{rw}$, $d_{cw}$, $d_{sw}$, and $d_{gb}$ are the road width, carriageway width, sidewalk width, and green belt width, respectively; *bHasTrees and bHasLamps* indicate whether the road segment has street trees and street lamps, respectively; and *strFilePath* and *strLampFilefile* are the file path of the 3D street tree model and the 3D street lamp model, respectively. A typical road template is shown in Figure 5. $C_1$ and $C_2$ are the start point and endpoint of the centerline, respectively; $T_1$ to $T_4$ are the polygon vertices of the carriageway, which can be obtained by the carriageway width ($d_{cw}$) and the centerline. Similarly, $B_1$ to $B_4$ and $S_1$ to $S_4$ are polygon vertices of the sidewalk and green belt, respectively, and can be calculated using the sidewalk width ($d_{sw}$) and green belt width ($d_{gb}$), respectively.

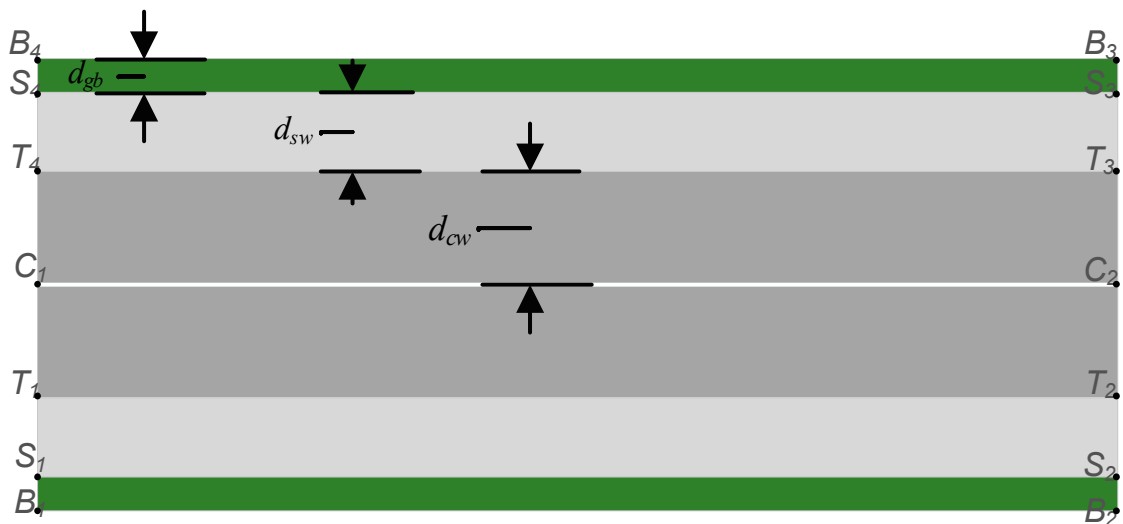

**Figure 5.** An example of the road template.

### 3.4.3. Interactive Building Design

As shown in Figure 4, the building distribution of a spatial plan can be created by simply dragging building templates from the building template library into the 3D scene. A building template has several attributes, as follows:

$$\text{Building Template} = \{na, tn, bm, lt, xs, ys, zs, xr, yr, zr\} \qquad (2)$$

where *na* is the name of the building template; *tn* and *bm* are the file path of thumbnail and building model, respectively; *lt* is the associated land use type, which is used to set the land use type attribute of a land parcel when a building template is dragged into the scene; *xs, ys,* and *zs are* the scaling factors in the x-, y-, and z-axis, respectively; and *xr, yr,* and *zr* are the rotation angles around the x-, y-, and z-axis, respectively.

As shown in Figure 4, a new building will automatically be generated according to the attributes of the selected building template after dragging a building template into the 3D scene. The land use type of the land parcel that contains the new building will be set to that of the building if it has not yet been set. Buildings on a land parcel should be the same type in a spatial plan. Therefore, further processes are needed if the land use type of the land parcel conflicts with that of the building when a building

template is dragged onto a land parcel. Normally, the user can remove the conflicted building using the building editing tool or divide the land parcel into smaller ones using roads or auxiliary lines.

### 3.4.4. Interactive Land Use Design

The boundary of a land parcel consists of road segments or auxiliary lines. Therefore, as shown in Figure 4, the geometry entity of a land parcel can be automatically generated or updated when creating or editing road segments and auxiliary lines. The land use type of a land parcel can be set to that of the building template that is dragged into it or to that of the land use template that is dragged onto it. The land use template defines the land use type and the appearance of a land parcel and can be described as follows:

$$\text{Land use template} = \{na, lt, fm, fc, ft\} \tag{3}$$

where *na* is the name of the land use template, *lt* is the land use type, and *fm* is the filling model, including filling with the color indicated by *fc* or filling with texture indicated by *ft*.

Like the road template and building template, land use templates are classified by land use type and stored in the land use template library. The user can create customized land use templates or use predefined ones.

### 3.5. Scientific Analysis and Evaluation Tools

Spatial planning is a comprehensive task that involves a wide range of subjects and requires scientific analysis from different disciplines. Therefore, spatial planners need to objectively analyze or evaluate spatial plan scenarios during the planning process, which is usually performed using tools from different software programs and is time consuming. However, in participatory spatial planning, planners need to respond to suggestions from participants in real-time during on-site discussions. Therefore, real-time scientific analysis and evaluation tools are essential for rural participatory planning.

A series of scientific analysis and evaluation tools for spatial planning were constructed on the 3D GIS platform, including the skyline analysis tool, the landscape pattern analysis tool, the ecological service analysis tool, the land balance analysis tool, the service radius analysis tool, etc. The skyline analysis tool helps the planner to evaluate the visual experience of the landscape from a 360° view at a point in the 3D environment using the skyline curve (vertical projection of the skyline), the skyline range (horizontal projection of the skyline), and the landscape hierarchy curve [9]. The landscape pattern analysis tool is used to analyze the landscape pattern using landscape pattern indexes, which are calculated based on land use data, including Shannon's diversity index (SHDI), the modified Simpson's diversity index (MSIDI), Shannon's evenness index (SHEI), patch richness (PR), patch density (PD), landscape shape index (LSI), etc. The ecological service analysis tool calculates the ecosystem service values based on land use data. Land balance analysis tool statistics indicate areas of different land types according to demands from different government sectors, which is helpful for generating the final report and judging whether the planned scenario is consistent with policies. Service radius analysis is useful for spatial location selection by calculating the service range of a facility based on the road network. More analysis tools can be developed using this framework.

### 3.6. Organization Tools for the Participatory Rural Planning Project

As shown in Figure 6, the data of the participatory rural planning project were organized in a tree structure using an XML-based data model. The root node is the project node that contains the basic information of the project, including project name, descriptions, location, initial view, etc. By double-clicking a project node, the system proceeds to the initial view stored in the project node. A project node has one or more child nodes corresponding to different planning schemes. Each planning node consists of several data layer nodes, including the boundary layer node, the road and auxiliary line layer node, the building layer node, the land use layer node, and the mark layer node. The mark layer includes point marks or polygon marks, which are used to document key information during

the interactive planning process to facilitate communication among stakeholders. All nodes can be switched on or off by changing the status of the markers in front of the node.

Marking and highlighting are extremely useful for smoothing and improving the interactions among participants during the on-site discussion process. A marking tool, which can be used to add and edit point-type marks or polygon-type marks, can be used to mark important suggestions, problems, conflicts, key notes, etc., during the discussion. A highlighting tool can be used to highlight spatial objects mentioned during the discussion to help planners and participants clearly express their opinions.

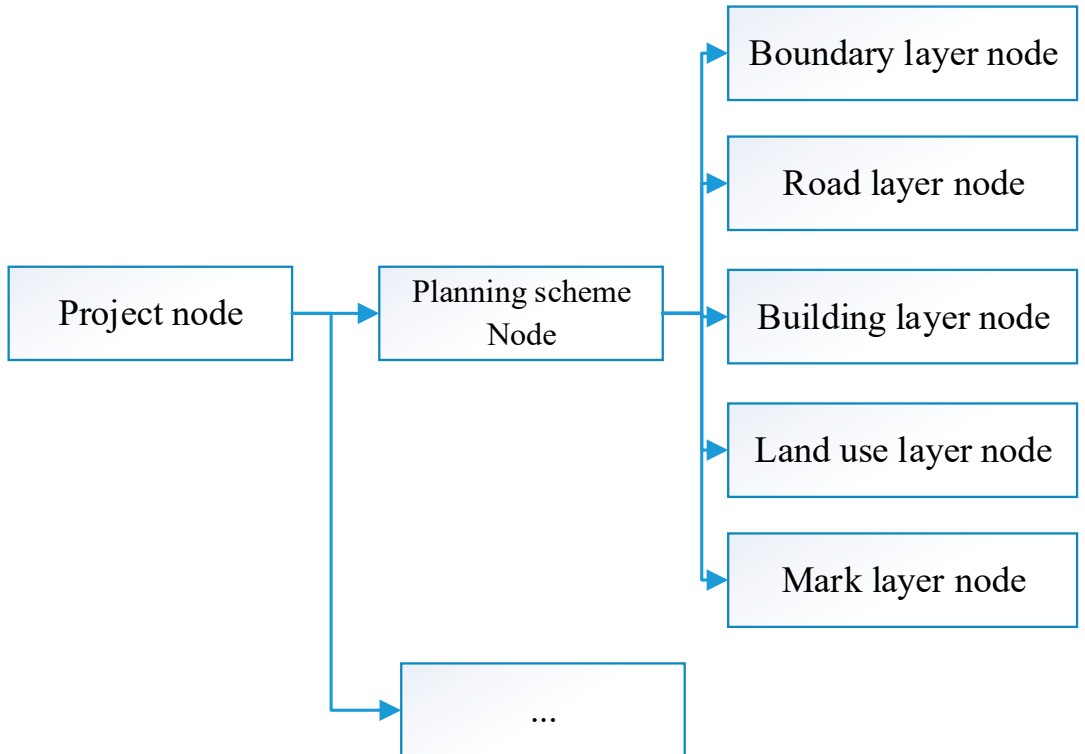

**Figure 6.** The data organization structure for the participatory rural planning project.

## 4. Experimental System and Case Study

An experimental 3D PGIS for participatory rural planning, named XYEarth, was developed based on the 3D virtual earth platform developed by Yu et al. [45]. XiaFan, NingHai, Ningbo, China, was used as a case study to test the feasibility of this system for rural participatory planning.

XiaFan, 16 km southwest of NingHai, consists of 168 households with a total population of 476, among which 28.8% of them are over 60 years of age. The village has about 10 ha of residential land, 15.2 ha of cultivated land, and 74.3 ha of mountain forest land. Of the land, 3.26 ha of collective land has been approved as industrial land. Residential land in the village is sufficient for villagers' needs, and a large number of older residences have been left empty by owners.

The major objective of this planning project is to improve the livable environment through village renewal through improving the village's infrastructure, public service facilities, villagers' living conditions, and environmental sanitation, etc. At present, one of the main obstacles to the development of XiaFan village is that some vacant old houses are hard to reclaim and transfer for village development due to their owners' calculation of value. In addition, due to historical reasons, some issues regarding land that is being illegally used must be solved through negotiations with villagers. As such, a 3D PGIS-based participatory planning method was applied in this spatial planning project to solve the above issues.

XYEarth was used to advance interactions among stakeholders including local government, local villagers, and planners in a typical participatory planning workflow (Figure 1). In Phase 1, basic data and information about the village were collected, including basic geographic data, previous spatial planning, planning at town level or above, natural and human resources data, social and economic development data, etc. In this phase, XYEarth acted as an information collection and visualization tool, supporting the creation of the present and historical 3D landscape of the village, providing the base for further planning. As shown in Figure 7, local villagers' participation in this data collection process was high with the help of the basic 3D scene that was created by the preinstalled remote-sensing images and a DEM. The present 3D landscape map (2019) of XiaFan was first created based on remote-sensing images and field surveys using the interactive landscape design tools of XYEarth by the planners. On this basis, with the participation and support of local villagers, the historical landscape maps of 1955, 1985, and 1995 were restored according to the memories of local elderly villagers.

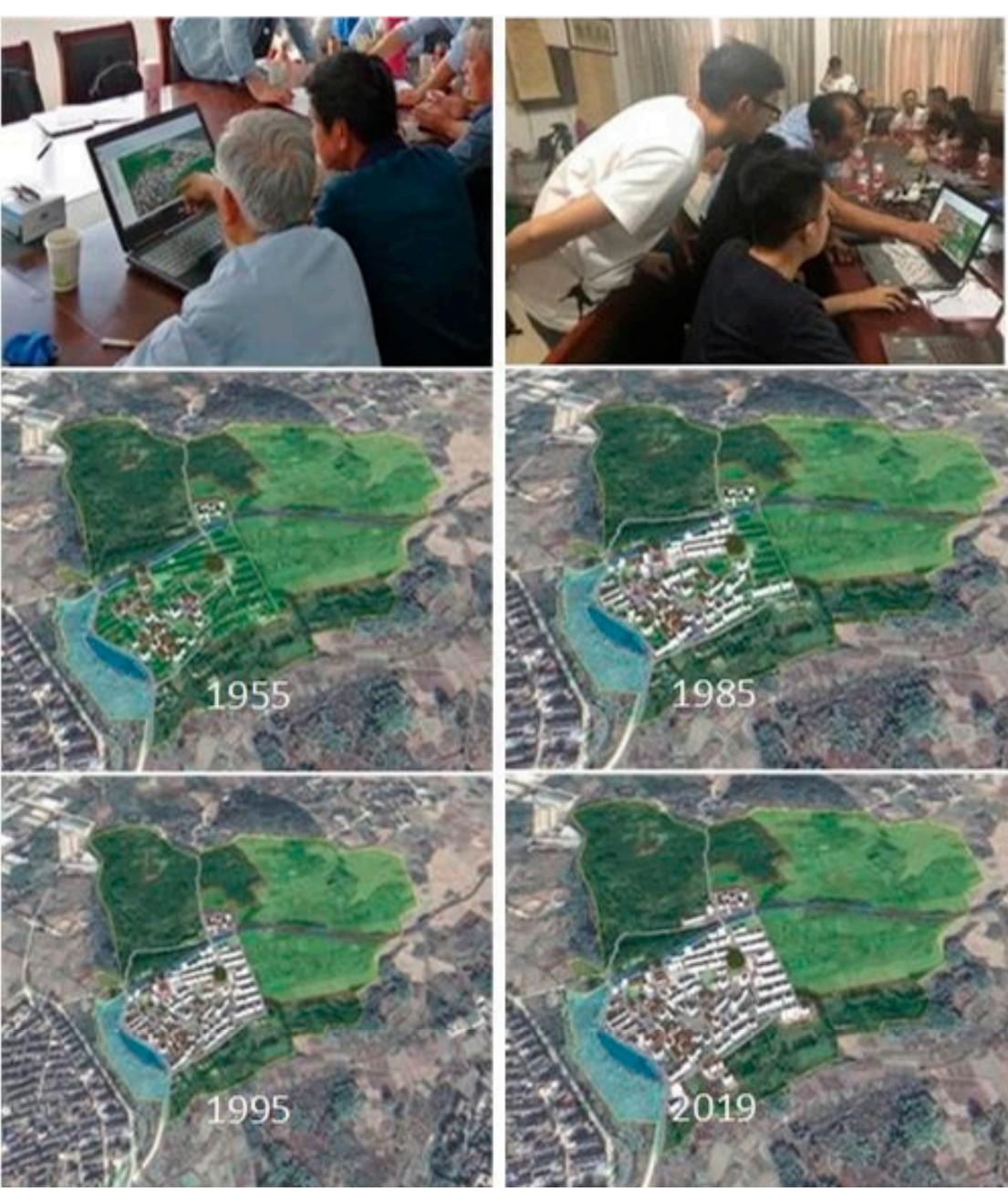

**Figure 7.** Data collection and present status analysis using XYEarth during phases 1 and 2.

The major task of Phase 2 was present status analysis, including problem identification and demand analysis, with the objective of preparing the necessary materials for future discussion in the next phase. In this phase, the marking and highlighting tools of XYEarth were mainly used. As shown in Figure 8, based on the present landscape, problems, demands, and conflicts from the local government and villagers were identified and recorded by interactively surveying and interviewing the stakeholders using XYEarth.

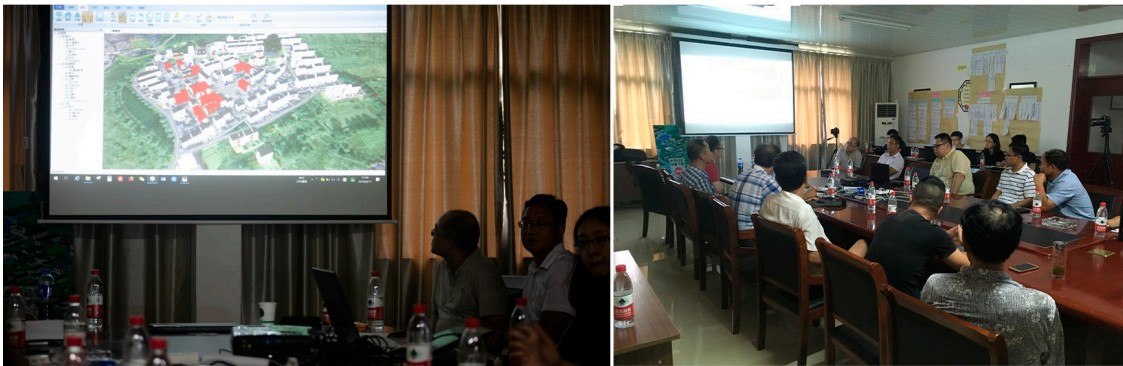

**Figure 8.** Conflict resolving based on on-site discussion using XYEarth.

In Phase 3, interactive planning and design were implemented. The XYEarth acted as an interactive communication platform through which various stakeholders could directly communicate in the 3D environment. In this phase, planners organized the interactive discussion, solved conflicts, and created an optimized plan using XYEarth, and participants, mainly local government and villagers, provided their demands and suggestions in this planning process, as shown in Figure 7. Interactive landscape design tools, scientific analysis and evaluation tools, and marking and highlighting tools were mainly used in this phase. For instance, in the process of site selection for public spaces in XiaFan, service range analysis of current public spaces (Figure 9a) and space syntax analysis of the current road network (Figure 9b) were conducted using the scientific analysis tools in XYEarth to provide references for further discussion. Site candidates for new public spaces proposed by planners and local villagers were marked (Figure 9c) and evaluated in several rounds of discussions in XYEarth (Figure 9d). Finally, new public spaces (Figure 8f) were designed based on the suitable sites (Figure 9e) that were identified.

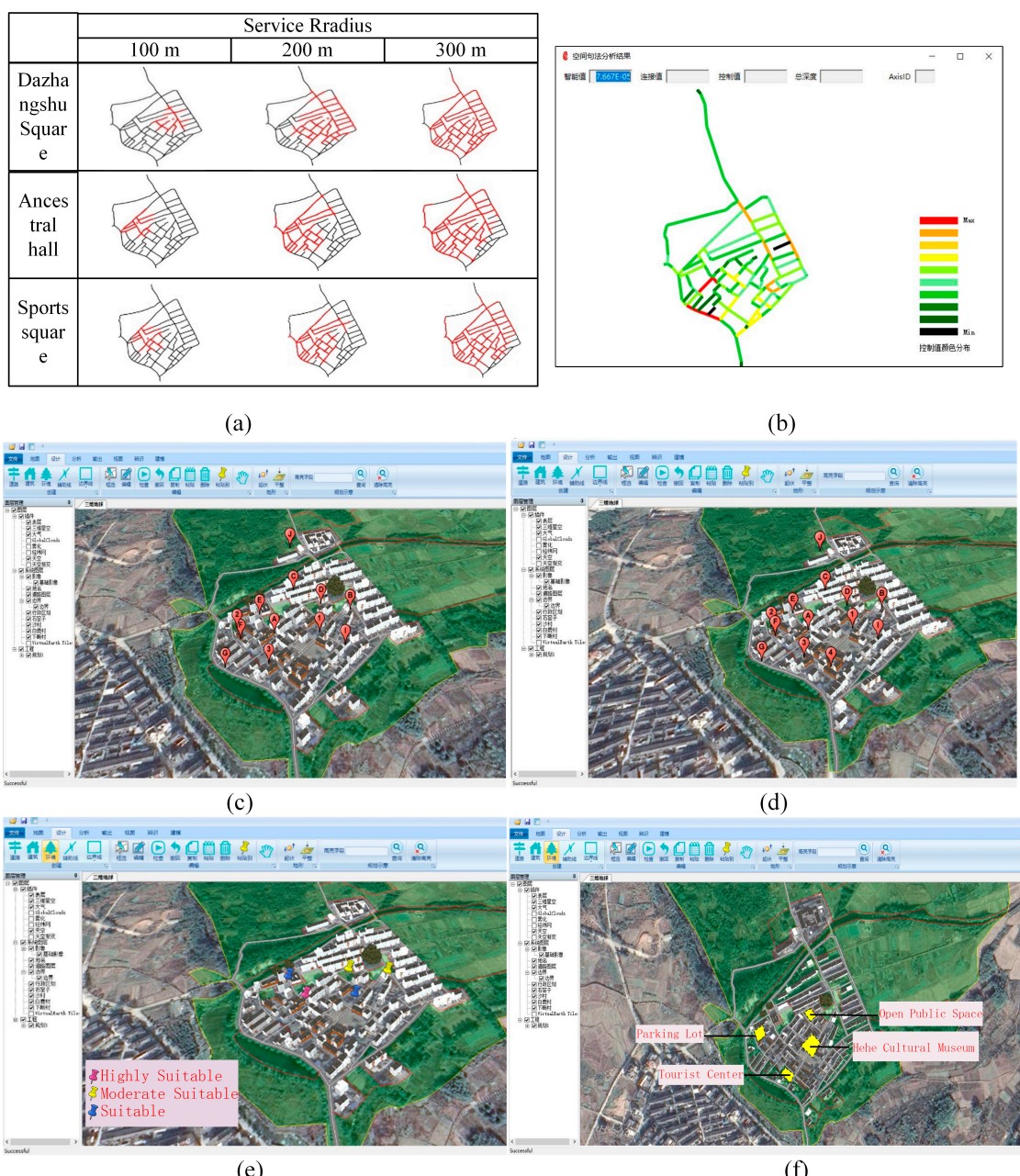

**Figure 9.** Site selection of public space using the scientific analysis and marking tools in XYEarth. (**a**) Service range analysis of current public spaces; (**b**) space syntax analysis of current road network; (**c**) initial candidate sites for new public spaces; (**d**) candidate sites for new public spaces after two rounds of discussion; (**e**) final candidate sites; (**f**) final design of public spaces.

## 5. Discussion

A high-quality geosolution should be user-centered [43]. The virtual globe provides a suitable platform for developing 3D PPGISs, not only because of its visualization capability of 2D and 3D spatial data on various scales but also due to the extensibility of developing and integrating application functions. Unlike web-based PPGISs designed to collect data from the public, the main objective of our solutions focuses on the requirements from on-site participants' interactions in rural spatial planning. Therefore, specific tools, such as interactive landscape design tools, marking and highlighting tools, and scientific analysis tools, were developed or integrated into this 3D PGIS.

Data collection is one of the basic tasks of rural spatial planning. Particularly, the collection of local knowledge is the key to the success of rural spatial planning. Local villagers are the major source of local knowledge. Web-based PPGISs have been widely used for public information collection (for example, Fast and Rinner [46] and Gomez-Barron et al. [47] and McLAREN [48]). However, local villagers usually have limited computer knowledge and are seldom able to skillfully use these professional tools. Our application experience shows that they prefer live communication, which therefore requires a platform supporting on-site information collection and interaction.

This visualization, as an effective method to provide material for human analysis and reasoning, is critical to facilitating decision making [49–51]. Three-dimensional (3D) visualization has been proven to be useful in piquing participants' interest and in presenting their opinions. However, this does not mean that a PGIS needs to strive for the best 3D visualization, especially in the context of rural spatial planning. Experience from our case study showed that on-site interactions do not require high-quality 3D visualization for rural spatial planning. Therefore, operational efficiency and visualization need to be balanced when developing a 3D PPGIS.

A PPGIS must now consider specific contexts, stakeholders, and other actors [19,44]. In the context of rural spatial planning, planners and local villagers are the major participants. Planners need to quickly respond to local villagers' suggestions and demands during on-site discussions. Therefore, to smooth and promote their interactions and communications on-site, ease of use is extremely important to a PPGIS. Our experience shows that planners need a PPGIS like a computer game with which they can quickly present both their and local villagers' opinions in the 3D map using only a few simple operations.

One of the roles played by the PPGIS is transparently and objectively making complex decisions [19]. It is important to allow participants to dynamically interact with the input and analyze alternatives in rural spatial planning. A PPGIS integrated with scientific analysis tools allows planners to evaluate plan scenarios in time, which can significantly shorten the rural spatial planning process.

## 6. Conclusions

To meet the demands for on-site interactions among various stakeholders during rural spatial planning, a virtual globe-based architecture for developing a new type of PGIS for rural planning was proposed in this paper. Based on this architecture, a virtual globe-based 3D PGIS, named XYEarth, was developed and applied in the participatory spatial planning of XiaFan village, Ningbo, China. This virtual globe-based system provides an interactive 3D communication platform for public participation in rural planning through the interactive design of landscape elements, real-time analysis of planning scenarios using scientific analysis and evaluation methods from different disciplines, and visualization of rural planning projects on any scale and from any viewpoint. Various stakeholders, even local villagers who have little knowledge of computers and planning, can rapidly and smoothly participate in the interactive on-site planning process with the help of this system. Although the 3D view was found to be more intuitive and attractive to local villagers than a 2D map, our findings suggest that the quality of the 3D visualization is not their focus. Our study also showed that the ease of use of the 3D PGIS is one of the major concerns of planners as they need to quickly present participants' demands and suggestions in the system during on-site discussions. The highlighting and marking tools were found to be quite useful for supporting the on-site discussions. Finally, although the scientific tools are important to planners for evaluating plan scenarios, they are rarely used during on-site discussions.

A new challenge that must be addressed in the future, however, arises from the application experience. As we focused our attention in this study on on-site interactions and communication among various stakeholders, the XYEarth system at present lacks the capabilities to support the remote participation of those stakeholders who cannot attend the face-to-face interviews and discussions. Professionals in various fields may provide valuable suggestions from different disciplines. However, a rural planning project usually does not have professors or experts attend the on-site discussion due

to the project's limited finance budget or time availability. Therefore, an extension of XYEarth with new features that allow stakeholders to participate remotely will be the focus of our future work.

**Author Contributions:** Conceptualization, Linjun Yu and Xiaotong Zhang; methodology, Linjun Yu and Xiaotong Zhang; software, Linjun Yu; validation, Feng He, Yalan Liu and Dacheng Wang; writing—original draft preparation, Linjun Yu; writing—review and editing, Yalan Liu and Feng He; visualization, Linjun Yu; project administration, Xiaotong Zhang; funding acquisition, Linjun Yu. All authors have read and agreed to the published version of the manuscript.

**Funding:** This research was funded by "Digital Simulation and Evaluation Model for Reconstruction of Town and Village Settlements Space" of the National Key Research and Development Program of China, grant number 2018YFD1100305.

**Acknowledgments:** We thank the National Key Research and Development Program of China for financial support. Special thanks are given to the local villagers of Xiafan Village, Ningbo City, China for their participations. Finally, the authors acknowledge the contribution of anonymous reviewers.

**Conflicts of Interest:** The authors declare no conflict of interest.

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
