# Peer review of "Participatory Rural Spatial Planning Based on a Virtual Globe-Based 3D PGIS"

_ijgi, doi:10.3390/ijgi9120763_

Round 1

Reviewer 1 Report

The paper covers the problem of rural spatial planning in China. The paper is informative; the introduction and literature review provide sufficient background and include sufficient references. The overall significance of the work is average.

There are some comments to the paper.

  1. Some journal policy requirements were not followed, namely: in abstract authors did not follow the style of structured abstracts.
  2. There are no any research hypotheses in introduction.
  3. Figures 1, 2, 3, 8 are not of high quality: written text in the figures is not so easy to read.
  4. Table 1: Authors should correct ‘Planning phrases’ to ‘Planning phases’; There are no any ‘Planner’ and ‘Major Result’ for phase 4; table design is done poorly.

Generally, after revision, the paper can be accepted for publication.

Author Response

Dear reviewer,

        Thank you very much for your comments concerning our manuscript entitled “Participatory Rural Spatial Planning based on a Virtual Globe-based 3D PGIS”. Those comments are all valuable and very helpful for revising and improving our paper. We have studied your comments carefully and have made corrections that we hope will meet with your approval. The revised portions are marked in Red. The primary corrections in the paper and the response to your comments are in the attached file.

With thanks

Best regards

Linjun Yu, XiaoTong,zhang, Feng He, Yalan Liu, Dacheng Wang

Reviewer 2 Report

General points: In this paper, the authors “proposed a 3D virtual globe-based PGIS aiming” to meet the requirements of a rural PP initiative in China. The context of “rural places” is very interesting and innovative for IJGI’s community. The article is readable, however, it would be nice to have a full language revision after rewriting. I must recommend improving the figures because they are poorly designed (especially figures 2,3,4,5,8). There is a method - that is not clearly announced - and interesting results. The authors also presented conclusions that are not conclusions: it revises the main outputs they found, but it should be more than that, it should really conclude what were the gains from the results? What are the real impacts – for the community - of developing such technology? And so on. The overall content of this paper is acceptable, however, the scientific dimension must be improved, as well as the sequence of developing such geoinformation solution. This last improvement would become this paper reproducible. In the following lines, I present some possible ways to enhance the approach of presenting the content.

Specific points: the introduction presents the gap of knowledge/technology that the authors are working with; a positive point to the community. I think the authors should define PPGIS properly, and, then, I recommend to read and use the content presented by Sieber (2006). Additionally, when presenting the Visualization/Design subjects, I recommend citing the classical literature covering these topics (see MachEachren’s papers/books; and more recent papers published by the ICA commissions on Cognitive Visualization, and Use and Users Issues). It would be valuable to insert some research questions delivered by the technological gap treated in this paper. That last consideration would improve – consistently - the scientific dimension of the discussion. By accepting these suggestions, I think the core theoretical subjects would be covered.

The method is poorly presented: it does not follow a reproducible sequence. That means, the reader would not be able to use this as a reference, once it brings no detail about what really matters for designing a geoinformation product that meets such interesting map use context. I recommend reading the research agenda produced by Griffin et al (2017). The authors should also consult methods for eliciting the requirements of users of geoinformation products (see Sluter et al, 2017); and classical methods of testing map users (see Suchan and Brewer, 2000). Please, pay special attention to Sluter, van Elzakker, and Ivanová (2017). It contains the core assumptions for designing geoinformation products under the Requirement Engineering context. The authors should present – in detail – the questions made during the sessions as well as the goal of each one of the questions. Without the questions, we cannot see if the interview sessions were representative in terms of data collection (and valid results). I must recommend rewriting the entire method, following a logical sequence. The authors should include how they would evaluate the results.

The authors should discuss the results based on the literature. It did not happen in the entire text. This is a very important gap the authors should improve. I think the best and suitable way to discuss this geoinformation solution result is to adopt the ISO (19157) point of view: is it “fitness for use?” “What does cover the Universe of Discourse?” “What are the visual variables that really matter while designing the solution?” “What differs the solution from another?” “What are the gains for the visualization process?” and many other possibilities.

Finally, I cannot recommend accepting this paper at this stage. The authors should observe – carefully - the improvements into the theoretical subject I suggested, which should reflect on the discussions.

As mentioned before, the literature covers the topic, however, I would like to share some of those papers I think could improve the literature review/method/discussion. The following research works must be considered:

FAST, V.; RINNER, C. Toward a participatory VGI methodology : crowdsourcing information on regional food assets Toward a participatory VGI methodology : crowdsourcing information on regional food assets. International Journal of Geographical Information Science, v. 32, n. 11, p. 2209–2224, 2018.

GÓMEZ-BARRÓN, J.-P.; MANSO-CALLEJO, M.-Á.; ALCARRIA, R.; ITURRIOZ, T. Volunteered Geographic Information System Design: Project and Participation Guidelines. ISPRS International Journal of Geo-Information, v. 5, n. 7, p. 108, 2016.

GRIFFIN, A. L.; WHITE, T.; FISH, C.; et al. Designing across map use contexts: a research agenda. International Journal of Cartography, v. 3, n. sup1, p. 90–114, 2017.

McLAREN, R.. Crowdsourcing support of land administration: a new, collaborative partnership between citizens and land professionals. Royal Institution of Chartered Surveyors (RICS) Report, November, p. 32, 2011

SHI, Y.; JI, S.; SHAO, X.; et al. Framework of SAGI Agriculture Remote Sensing and Its Perspectives in Supporting National Food Security. Journal of Integrative Agriculture, v. 13, n. 7, p. 1443–1450, 2014.

SIEBER, R. (2006) ‘Public Participation Geographic Information Systems: A Literature Review and Framework’, Annals of the Association of American Geographers.  Taylor & Francis Group , 96(3), pp. 491–507. doi: 10.1111/j.1467-8306.2006.00702.x.

SLUTER, C. R., VAN ELZAKKER, C. P. J. M. AND IVÁNOVÁ, I. (2017) ‘Requirements Elicitation for Geo-information Solutions’, The Cartographic Journal. Taylor & Francis, 54(1), pp. 77–90. doi: 10.1179/1743277414Y.0000000092.

SUCHAN, T. A. AND BREWER, C. A. (2000) ‘Qualitative Methods for Research on Mapmaking and Map Use’, The Professional Geographer. John Wiley & Sons, Ltd (10.1111), 52(1), pp. 145–154. doi: 10.1111/0033-0124.00212.

YAN, Y.; FENG, C.-C.; CHANG, K. Towards Enhancing Integrated Pest Management Based on Volunteered Geographic Information. ISPRS International Journal of Geo-Information, v. 6, n. 7, p. 224, 2017.

YU, Q.; SHI, Y.; TANG, H.; et al. eFarm: A Tool for Better Observing Agricultural Land Systems. Sensors, v. 17, n. 3, 2017.

Author Response

(The authors gave the same response as above.)

Reviewer 3 Report

Dear authors, I find your manuscript very interesting and the participatory GIS is indeed a very important issue that can improve real public participation in the planning process and decision-making.

I've found your software solution very nice and clean, having only the important features that are needed and on a path of improvement over time. Although 3D PGIS (or Participatory 3D Modelling) is not new, and we have seen similar solutions in the last decade (i.e. http://pgis.cta.int/en/index.html, https://www.mdpi.com/2220-9964/8/6/253/pdf), I think that your implementation for urban planning purposes enhances the effective participation of the general public. I think that your work is worthy of publication, after addressing the comments below and after complementing the manuscript with an assessment of the tool usability, i.e., which software features worked as expected and which did not? Were participants engaged with the process? Did it helped to foster participation? Did participants asked for different software features to better understand the proposed changes in the spatial plan? (see the work of Lafrance et al., 2019 for an example of this kind of discussion).

Please see specific comments below that are aimed to improve the manuscript.

--- Abstract ---
Ok

--- 1. Introduction ---

47-48: International readers could not be familiar with the size of the Chinese small towns and villages. It might be useful to briefly define in terms of population size how big or small are the Chinese small-towns and villages.

57: ecology-orientated and economical land use-orientated? These term does not seem to be written in proper English. Terms often used in the academic literature are ecological planning or ecology-oriented planning, economic planning, place-based economic development, market-oriented planning, among others. Please verify and use proper terms for an international audience.

62-63: is not a mandatory... please rewrite: is not madatorily required by law, or put it simply: is not mandatory, or is not compulsory (mandatory and compulsory, both words mean required by law, so there is no need to state that again in the same sentence).

74: ...required by law in many countries. Could the authors list just a few? Are those developed countries or there are examples from low and middle income countries?

76: Zhang, et al 2019: I think that the proper form to cite this reference is: Zhang et al., 2019.

82-83: public participation trends to take... : I think that the verb "tends", instead of "trends", would be more appropriate in this case.

92: public participatory -> public participation.

--- 2. Related works ---
General comment: There is a work that should be cited in this section:
https://www.mdpi.com/2220-9964/8/6/253/pdf, as it mentions the need of this kind of tools for urban planning, and specifically discuss the advantages and drawbacks of 3D visualizations for this purpose.

115-116: participation planning -> participatory planning.

117-119: It seems that the manuscript lacks a word or punctuation sign in this sentence.

124-125: using simplified graphical user interface -> using a simplified...

134: I assume that VR stands for virtual reality, but I am not sure because it hasn't been defined before in the text. Please define each acronym the first time they appear in the manuscript.

138: to present planning scenarios for specific project -> for specific projects? (plural), please verify.

140: which -> that

148: Which of these softwares are open and which are not? which are applications and which are software development kits (SDK)?

154: face to face -> face-to-face

154: ...than online model... -> ...than the online model...

155: not unfamiliar? this seems like double negation, please correct: not familiar, or unfamiliar.

155: computer -> computers

156: Rural planning spatial planning -> rural spatial planning.

156-158: This sentence is hard to read. You say first that rural spatial planning is smaller and less complex than urban spatial planning. And then you mention a rural spatial planning requirement. I suggest to split the sentence into two different ones and clearly convey your message: 1- rural spatial planning is less complex than its urban counterpart, 2- however, rural spatial planning does require a platform that supports 3D landscape designing and decision making.

--- 3. Virtual globe-based 3D PGIS ---
175: Can you provide some references to examples? The reader will appreciate that.

186: The figure caption lacks the proper use of punctuation and capital letters.

200: ...of participatory... -> ...of the participatory...

202: The figure caption lacks the proper use of punctuation and capital letters.

203: serveries -> services

213: ...just like play a computer game. -> ...just like playing a computer game.

219: tool -> tools; realizing -> allowing the interactive...

221: The figure caption lacks proper use of punctuation and capital letters. Also in the figure: Templet is hardly used in American English, please use "template" for more clarity towards your readers.

229: is developed> or was developed?

233-235: This is a very nice feature of the system!

247: The figure caption lacks the proper use of punctuation and capital letters.

264: using building editing tool -> using the building editing tool.

275: User can... -> The user can...

287: ...and et al? What did you mean with "et al"? Please stated it simply or use a more common expression like "among others" or something similar.

288: helps planner -> helps the planner.

294: and et al?

295-296: Land balance analysis tool statistics areas of different land types according to demands from different government sectors. Please rewrite this sentence so it gets easier to understand... did you missed a word or something? (something like: The land balance analysis tool computes the area statistics of different land types...).

303: The root node is project node -> the root node is the project node...

313: The figure caption lacks the proper use of punctuation and capital letters.

--- 4. Experimental system and case study ---

325-326: please separate the figures from the units, i.e.: 10 ha, 15.2 ha, 74.3 ha, etc.

330: renew al -> renewal.

331: and et al?

336: to solve above issues -> to solve the above issues.

337: interactives -> interactions

340: The table caption lacks the proper use of punctuation. Also in table 1: Confliction resolving -> Conflict resolving.

354: The figure caption lacks the proper use of punctuation.

356-357: in next phrase -> in the next phase.

362: The figure caption lacks the proper use of punctuation.

373: was -> were (plural).

374-375: Please rewrite in an orderly manner. For example: The design of new public spaces was made based on the suitable sites that were found.

--- 5. Conclusion and future work ---

393: who has -> who have (plural); computer -> computers.

Author Response

(The authors gave the same response as above.)

Reviewer 4 Report

This is an interesting paper concerning public participation processes in rural planning, taking as case-study a village in China. The paper is highly informative as far as the 3D PGIS are concerned and deserves publication. Nevertheless, the manuscript is of rather poor quality. The text is a bit "chatty", there are many repetitions and this is evident from the very beginning ( please see abstract).

I have tried to shorten your abstract in order to show to you that your test  can be easily "scaled-down" without loosing its scientific soundness.

My suggestion ( 100 words less than yours) is the following :

"In current times of spatial planning reform in China, rural planning is becoming increasingly important. Although public participation is key of success for rural planning, it is difficult to engage local villagers and local government staff, in spatial planning projects due mainly to lack of planning knowledge and computer skills. Therefore, this paper discusses the development of a virtual globe-based 3D Participatory Geographic Information System (PGIS) and the approach of 3D PGIS- based rural spatial planning. A prototype of the above tool was developed to support public participation in the spatial planning process by providing interactive landscapes design, real-time scientific analysis of plans and planning data visualization and management. Experiences of applying this 3D PGIS based planning approach in XiaFan Village, Ningbo City, China demonstrates that  locals’ participatory capacity was highly promoted with their interests in 3D PGIS visualization being highly activated.The interactive landscape design tools allow stakeholders present own suggestions and designs like playing a computer game,  thus improving their interactive planning abilities on-site. The scientific analysis tools make planners analyze and evaluate planning scenarios in different disciplines on real-time to quickly response suggestions from participants on-site. Functions and tools such as data management, marking and highlighting were proved to be very useful to smooth the interactions among planners and participants. The paper concludes that virtual globe-based 3D PGIS highly supports participatory rural landscape planning processes and is potentially applicable to other regions."

I have also made some comments of minor importance on the text itself ( please see attached file). In general I would advice you to keep the classic structure Introduction/Materials and methods/ Results/ Discussion/Conclusions and restructure your manuscript taking also into account that many "cuts" are possible. 

Ι would like to see the new manuscript, after improvements, please. 

Author Response

(The authors gave the same response as above.)

Round 2

Reviewer 2 Report

I think now the paper is suitable for publication. There is reproducible method and a scientific discussion, allowing the readers to visualize possible ways to enhance similar PPGIS applications. I have only one consideration (line 495) on the citation “Sieber et al, 2017”; do you mean Griffin et al, 2017? Please, check this point out.

Author Response

Dear reviewer,

      Thank you very much for your comments concerning our manuscript. Those comments are all valuable and very helpful for revising and improving our paper. We have studied your comments carefully and have made corrections that we hope will meet with your approval. The revised portions are marked in Red. The primary corrections in the paper and the response to your comments are as follows:

 Comment 1:

    I think now the paper is suitable for publication. There is reproducible method and a scientific discussion, allowing the readers to visualize possible ways to enhance similar PPGIS applications. I have only one consideration (line 495) on the citation “Sieber et al, 2017”; do you mean Griffin et al, 2017? Please, check this point out.

 Response:

       Thank you. “Sieber et al, 2017” should be “Griffin et al, 2017”. This error has been corrected as follows:

    “A PPGIS must now consider specific contexts, stakeholders, and other actors (Sieber, 2006; Griffin et al., 2017).”

Thanks again.

Reviewer 3 Report

Dear authors,

I've seen that you addressed all my comments and provided a new discussion section on the usability topic as I recommended. I think that the work has been improved significantly and now will be ready for publication after you address minor typos or spelling errors as noted below.

Abstract:

L 18-19: ...in "the" success of...

1.Introduction

L 81: urban area -> urban areas

L 105, 106: has -> have (you spoke about "systems", in plural)

3. Materials and methods

L 254: caption of Figure 2. Please start the sentence with a capital letter. the -> The

L 366-367, L 399 and L 429: "and etc." This expression is hardly used in academic papers (first time I see it). I recommend changing it to ", etc." (without the "and").

5 - Discussion

L 498: import -> important; planner needs -> planners need

L 501: One of roles -> One of the roles

6 - Conclusion

L 527: arise -> arises

Author Response

Dear reviewer,

    Thank you very much for your comments concerning our manuscript. Those comments are all valuable and very helpful for revising and improving our paper. We have studied your comments carefully and have made corrections that we hope will meet with your approval. The revised portions are marked in Red. The primary corrections in the paper and the response to your comments are in the attached file. 
